# Medievals and Moderns in Conversation: Co-Designing Creative Futures for Underused Historic Churches in Rural Communities

Timothy J. Senior [1,*], Tom Metcalfe [2], Stuart McClean [3], Alexander Wilson [4], Simon Bowen [5], Marianne Ailes [6] and Ed McGregor [7]

1. Supersum Limited, 71–75 Shelton Street, Covent Garden, London WC2H 9JQ, UK
2. Centre for Innovation and Entrepreneurship, Richmond Building, University of Bristol, Bristol BS8 1LN, UK; tom.metcalfe@bristol.ac.uk
3. School of Health and Social Wellbeing, Frenchay Campus, Coldharbour Lane, University of the West of England, Bristol BS16 1QY, UK; stuart.mcclean@uwe.ac.uk
4. School of Architecture, Planning and Landscape, Newcastle University, Newcastle upon Tyne NE1 7RU, UK; alexander.wilson@ncl.ac.uk
5. Open Lab, School of Computing, Newcastle University, Newcastle upon Tyne NE4 5TG, UK; simon.bowen@ncl.ac.uk
6. School of Modern Languages, University of Bristol, 17 Woodland Rd., Bristol BS8 1TE, UK; marianne.ailes@bristol.ac.uk
7. The Churches Conservation Trust, St. Thomas' Church, Thomas Lane, Bristol BS1 6JG, UK; emcgregor@thecct.org.uk
* Correspondence: team@supersum.works

**Abstract:** For many living in rural areas, the loss of traditional community assets and increased social fragmentation are a common feature of everyday life. The empty village church is a poignant symbol of these challenges; yet, these are sites that hold considerable potential for new placemaking solutions that respond to the needs of communities today. This means looking beyond "the traditional village church" to recognise a longer history of church adaptation and resilience within the lives of communities. In this paper we ask: how can co-design, projected through a Wicked problems and Clumsy solutions lens, help imagine new futures for communities and their historic churches today? Clumsy solutions consider a plurality of different perspectives on the nature of problems and their resolution to deliver more effective solutions with broad appeal. In the search for clumsiness, we turn to 'long history' and 'slow technology' for inspiration, uncovering deeper resonance with historical communities of place and anchoring that continuity within church sites themselves. Our paper demonstrates how Wicked/Clumsy thinking can account for the challenges faced by rural communities today, bootstrap co-design activities in the development of clumsy solutions, and uncover clumsiness in long history and slow technology dimensions—together laying the foundation for new placemaking strategies.

**Keywords:** slow technology; internet of things; cultural heritage/history; storytelling; co-design

## 1. Introduction

The medieval English parish (a geographical unit of church governance) is an important point of origin for many communities in the UK today. Whilst parish structures retain a governance role, their significance in structuring and shaping community life has diminished dramatically. Today, populations in rural areas face increasing uncertainty in the search for vibrant community life; yet, the historic (often medieval) churches that are a common feature of rural parishes remain under-used. The uncertain future of rural communities and their churches today is a classic 'wicked' problem [1] in that it is unique, enduring, multi-faceted, hard to define, and subject to a plurality of views on the nature

of the problem and the best solution to it. Wicked problems demand so-called 'Clumsy' or 'poly-rational' solutions, those that creatively combine a plurality of viewpoints to find a solution with broad appeal [2]. It is here that our research interest emerges: How can co-design—worked through Wicked problems and Clumsy solutions thinking—help to imagine new futures for communities and their historic churches?

In the search for a Clumsy solution, we turn to two key sources of inspiration: 'long history' (as we will term it) and 'slow technology' [3]. We argue that taking a long history perspective on medieval churches can uncover rich themes that speak to contemporary life, so suggesting forms of continuity and resonance with historical communities of place. As such, it can generate invaluable resources for co-designing futures that reinvigorate rural communities and heritage assets. In parallel, the slow technology perspective can enable new forms of learning, engagement and reflection in place, anchoring those interactions (both physically and digitally) to medieval churches themselves. This paper reports on the development of a Clumsy placemaking strategy at the rural parish of Brockley in Somerset, UK, where our study involved co-design activities with parish residents in and around the medieval church of St. Nicholas—a church no longer in regular use and under the care of the Churches Conservation Trust (CCT).

Our paper makes three contributions: first, we show how Wicked/Clumsy thinking can account for the challenges faced by a rural community today, demonstrating how this can bootstrap co-design activities and set the trajectory for Clumsy solution development; second, we outline how clumsiness can be found in the long history and slow technology dimensions of this work, laying the foundation for a full Clumsy placemaking strategy; third, we suggest what Wicked/Clumsy thinking might offer co-design in terms of critically understanding 'the social', grounding 'design futures' in more than 'the present', and challenging a tendency towards linear thinking between problems and their solutions.

## 2. Background

### 2.1. Rural Uncertainty and Empty Churches

The medieval parish has played a central role in the formation of 'community' as we might still recognise it today. The parish embedded individuals into new types of horizontal and vertical social networks, was a generator of local infrastructure, business credit and education, and the principal driver behind sponsors of, and audiences for, creative and cultural activities [4]. Numbering over 12,000, the medieval parish has been described as the "micro-laboratory" of new political structures in the country at large, facilitating the transition from feudal to post-feudal society, and so setting the process for eventual democratisation and secularisation [5].

Today, rural parishes are as diverse as urban economies. In England, for example, rural areas house 17% of the population but 24% of all its registered businesses [6], with creative micros, small-scale manufacturing and non-profits (amongst others) active across a variety of sectors. That same transition from predominantly agricultural to modern economies, however, finds parallel in changing demographics, the loss of traditional community assets, increased social fragmentation, the rise of social isolation and the emergence of a wellbeing crisis. Social isolation, for example, is a significant and rising public health concern and associated with a range of health and wellbeing implications [7,8]. Living in rural areas can lead to a number of specific health risks that may stem from population changes (outward migration of younger people and inward migration of older people) and diminished access to services and infrastructure (with limited community support locally driving a rise in social exclusion and isolation). In itself, living in a remote rural area is an important risk factor both for social isolation and loneliness, as well as high morbidity and mortality rates in older age [9,10]. Overall, poorer outcomes for older people in rural localities have only intensified during the COVID-19 pandemic [11].

This challenge—a diminishing of the potential held by rural places to thrive—calls for new types of action if people are to live better together in their communities. There is now a wider recognition of the importance of placemaking strategies, those that are locally rooted,

accountable to communities, and work to the wider benefit of all. Increasingly, therefore, there is a turn to community-led initiatives, social enterprises and local businesses to address these issues and drive new forms of social capital [12]. In this drive, "third places" (such as pubs, village halls and parish churches)—places to come together, enact and sustain community wellbeing—will likely play a central role in placemaking strategies that can respond to the needs of communities today; third places can be seen as protective to health and wellbeing across the life course, driving resilience, providing buffers for wellbeing and loneliness, and helping us to think widely on the healthy relationships that can exist between the built environment and connected communities [13–15]. Whilst communities are increasingly identified as critical to the development of new placemaking strategies, however, the question of how best to empower communities to do so remains open.

The empty, historic parish church is emblematic of the uncertainty facing rural populations today, yet might form the kernel of a robust response to social challenges such as loneliness and social disconnection. Today, the *historic* role of the parish church at the centre of community life (encompassing, for example, social service, poor relief, education, creative and commercial activities in addition to its central purpose in Christian teaching [5,16,17]) may go largely unappreciated. Superficially, attitudes towards churches as prescriptive religious places (places of dedicated Christian worship) can be a stumbling block to wider conversations around church futures in their communities. The need for that conversation, however, is only becoming stronger, with dwindling congregations, an ageing demographic of churchgoers, and soaring repair and maintenance costs [18]. The wide-reaching 2017 Taylor report on the long-term survival of the Church of England's churches points to the urgent need for active churches and congregations to engage with a wider cross-section of communities, to drive broad grass-roots commitment, to be entrepreneurial in generating new uses and revenue streams, to pool insight, and to link into expert-led networks of community and repair action [19]. Thinking to the future of under-used medieval churches, therefore, addresses a double need: the reinvigoration of community life and the safeguarding of valuable heritage for future generations. This is an alignment with profound historical resonance and potential for scale. Of the 15,700 Church of England churches, 57% are in rural areas with 91% listed historic buildings [18]. At scale, we can see the potential for the rural parish to be a "micro-laboratory" for inclusive and vibrant communities once more—mirroring its historic role.

## 2.2. Co-Design with Faith Communities

The need to engage communities in the development of new, mutually beneficial futures for churches is now clear. Historically, however, investment has been earmarked largely for much-needed repair and maintenance work, with little attention paid to the social role of communities in safeguarding church futures [19]. That engagement is now critical, but there is no 'one size fits all' solution, with every church and community expressing different local considerations, values, needs, and resources [19]. Additionally, faith communities vary in their degree of openness towards wider community-use of places of worship [20], only further emphasising the importance of tailored local responses. Testament to this 'bigger picture' is the enormous variety of regenerative outcomes seen for churches today: from the preservation of buildings as heritage assets, the use of church sites by communities for the benefit of local residents and businesses (run as social enterprises or co-operatives), to the installation of facilities (such as heating, toilets, kitchenettes, meeting and office units) to create spaces more suitable for social and commercial purposes.

It is here, in developing new regenerative strategies with and by communities in response to local needs, that the value of co-design emerges. In this role, we focus on co-design's potential to engage a wide variety of people more meaningfully in the design of solutions to shared problems, a way to "elicit complementary knowledge and expertise and utilise their creative capacity" [21]. Particularly relevant in this current work is how co-design approaches can aid the disruption of conventional thinking, direct attention away from norms and unquestioned values, and offer new ways to inhabit the experiences

of others whilst emphasising the social nature of collaboration and its intended impacts or outcomes. Such approaches might be built around probes (to inspire, inform, drive participation or dialogue), toolkits (purposeful collections of material tools used in collaborative activities), prototypes, provotypes (provocative prototypes) and games. Each has a different affordance in supporting individuals to act creatively under their own discretion or actively participate in a facilitated process of reflection, discussion and exchange [21,22].

Co-design approaches are now being adopted by groups actively seeking new futures for historic places of worship, whether locally at a single site, through multi-partner university-led research projects or in terms of scalable policy instruments, e.g., [23–25]. The AHRC-funded 'Empowering Design Practices' research project (2014–2021), for example, has explored how community-led co-design can create "more open, vibrant and sustainable places that respect and enhance the heritage of their buildings" [24]: working with 55 communities of multiple faiths and denominations in England, this extensive project offered a variety of design and training workshops addressing central themes as documenting and evaluating activities, mapping local contexts, capturing views and opinions, and generating new ideas (some adopted in this paper). In parallel, the recently completed Taylor Review Pilot (2018–2020; delivered by the Department for Digital, Culture, Media and Sport and managed by Historic England) offers a first step towards a scalable policy instrument for sustainable church futures across England, simultaneously addressing issues of maintenance (through Fabric Support Officers) and community-led development (through Community Development Advisers, CDAs): working with 205 listed places of worship, CDAs helped communities understand their current situation, address barriers to change, develop new ideas and put action plans in place [25]. Here, the Process Map developed by CDAs is commensurate with, and lays the foundation for, an intensification of co-design activities at scale. Whilst co-design is gaining recognition in this field, there are still important questions to be asked around how these approaches might be best theorised, designed and programmed in response to community needs.

### 2.3. Wicked Problems and Clumsy Solutions

In this paper, we explore a novel Wicked problems/Clumsy solutions methodology to co-design. It is an approach, we argue, that can help better understand the nature of rural uncertainty today and point towards innovative forms of placemaking. Wicked problems are particularly difficult to resolve as they are "unique, costly, enduring, multi-faceted and hard to define, involve lots of people and organisations advocating a plurality of views, and cut across a number of social and natural domains" [26]. Wicked problems can be contrasted with 'tame' problems, those that can be addressed through a linear decision-making process starting with unambiguously defining the problem, collecting and analysing relevant data, and then delivering a clear and measurable solution [1]. Wicked problems have no definitive formulation: there are multiple ways to characterise them and define potential responses (a problem-solution nexus that is technically limitless). Every wicked problem is also essentially unique, symptomatic of other problems, and embedded in wider contexts that themselves change over time. As such, there can be no formal process to determine how correct a solution is, only varying judgements of "good or bad" (and no ultimate test of a solution because its full impacts will always be hard to ascertain). It follows that wicked problems have no stopping rule because solutions will not only further uncover the problem but change the formulation of the problem itself. Finally, each iteration of a solution is a "one-shot operation" with every attempt counting significantly, and with those leading an intervention having "no right to be wrong".

A detailed analysis of 'church futures' as a wicked problem is beyond the scope of this paper, so we instead provide a sketch: Whilst the relationship between churches and their communities has changed historically, there have always been multiple stakeholders (incumbent congregations, local residents, philanthropists, Parochial Church Councils, local and central government etc.) who differ in their understanding of the roles and responsibilities connecting communities to their churches. Each church has its own unique

character, one holding multiple, often competing, value narratives at any given time (religious, spiritual, aesthetic, cultural, social, economic) and intersecting with wider local, regional and national contexts (for example in terms of demographic and economic change). At this critical junction in the future of communities and their churches, all interventions must count significantly. There is no 'magic bullet' to securing the future of historic churches, but rather multiple possible solutions, each anticipating a different future down the line. It is significant that the focus on top-down investment in church futures in recent decades (coupled with an elegant hierarchical position that 'communities will just do their bit') has not helped build sustainable church futures; more 'market-driven' solutions, such as placing the operation of a church building in the hands of a business or social enterprise with a specific target audience or market sector in mind, may be an effective route to secure the immediate future of a church building but at the cost of a broader and more inclusive community role; finally, community-led solutions are prone to the challenges typical of grassroots movements (such as difficulties in broadening participation or securing funding).

Wicked problems call for a certain type of solution, termed 'Clumsy' (or poly-rational). Clumsy solutions are so-called not because they're unreliable or unstable, but because they consider a plurality of perspectives and rationales (from different strategies such as those above) to find more effective shared solutions with broad appeal. A key position of clumsiness is that it should produce more effective outcomes than 'elegant' solutions can offer (those optimised around just one of definition of the problem), whilst still appealing to those stakeholders.

Befitting 'wicked' characteristics, the process of developing Clumsy solutions must remain open-minded to the causes of problems and possible solutions, work closely and broadly with extended peer communities, and be embedded deeply into the unique situations where problems arise (whilst also seeking to understand broader contexts, causes, social movements and patterns of change). Clumsy solutions, themselves, must not only be pluralist in nature but pluralist in *the right way* and lay the grounds for their critique and renewal in response to their own impact on the problems they tackle. Wicked problems and Clumsy solutions thinking, therefore, has implications for how you go about understanding communities (and the nature of pluralist solutions), design collaborative activities with communities, and parameterise the clumsy nature of interventions. Whilst the value of design-led methods in the development of Clumsy solutions has been established [27], in this paper we will argue that Wicked/Clumsy thinking can, in turn, benefit the co-design field by critically foregrounding 'the social' in 'community' and challenging a tendency towards linear thinking between problems and solutions in the social space.

*2.4. Expanding the Clumsy Solution Space: Long History and Slow Technology*

If the search for Clumsy solutions is only strengthened by looking beyond conventional explanations of a given problem and its likely solution, then an expansion of the 'possible solution space'—the space in which ideas are generated and explored—is highly desirable. In our work, we introduce two key dimensions to expand this space: long history and slow technology.

The first introduces into the search for clumsiness a much longer history of church adaptation, resilience and purposeful iteration than often engaged with at present. When we take the long view of historic church sites and their communities we encounter a wider spectrum of relationships between religious, spiritual, social, commercial and secular life and the social constructs that generate them. By turning to a site's own long history, we open-up a greater variation in accounts of viable (and non-viable) social life. This can aid the search for solutions that lie beyond habitual responses to long-standing (Wicked) problems, grounding the problem and its possible resolution in the rich historical experience of communities themselves. Not only is it important when re-imagining the place of a historic building in society to "understand a building's past as embedded in architectural and artistic objects and features as well as in people's memories, rituals and cultural associations and traditions" when developing future interventions [28] but also to recognise how the

past can be a source of alternative models for framing present challenges and possible futures. That "buildings and their meanings [are] ever-changing, negotiated, re-interpreted and adapted in relation to their wider historic environment and changing social and cultural norms and values" [28] has always been the case: rather than "a crisis of the now" (subject to forward-looking design solutions), engaging with a site's long history can change how we think about the time-stamp, temporal-extent and changing nature a problem and its possible solution.

The second dimension we introduce into the search for clumsiness is the potential of digital technology embedded into the everyday social and spatial experience of communities. At face value, digital technologies seem well placed to help connect people and facilitate exchange within communities. There have been well-documented and sustained efforts to use technology in this way, for example to engage people in community decision-making and to provide input on matters that impact people's lives [29]. However, a focus on expedient approaches to engagement, typically on screen-based personal internet-connected devices (e.g., smart phones) has come to dominate the field, with a focus on increasing ease, simplicity, speed and efficiency of technology-use to encourage participation, rather than exploring forms of engagement that could be more meaningful, expressive, reflective, engaging and situated for intended audiences [30,31]. Indeed, typically designed to facilitate quick-engagement on personal internet-connected devices, standard communication technologies offer little scope to connect with the material, social and cultural lives of communities in the challenges they face [29,32]. The benefits of more deliberative and engaged approaches (particularly non-digital), and the affordances of digital-physical technologies that inspire engagement, discussion and reflection are well documented, but rarely applied in practice. In this work, we engaged 'slow technology' practices, an approach to technology design that encourages "learning, understanding and presence to give people time to think and reflect"; this is technology designed not to compress time (to do tasks quicker), but rather supply time to do new things: time productive, not time consuming [3]. Slow technology can open-up different conversational modes (formal and informal), introduce conditions for thoughtful and respectful public contribution-making, and be made responsive (in design) to the unique characteristics of a place and its community [3]. In the search for clumsiness, we argue that a slow technology dimension that is shared, tangible and in-place may change how a church site can be foregrounded in community life: we ask how it might drive and expand participation beyond conventional digital channels, make the church 'visible' through physically placing it at the centre of a dynamic digital exchange within the community, and foster a connection (through design) between historical and present worlds—foregrounding historical continuity as part of an expanded clumsy solution space.

## 3. Methods

### 3.1. Co-Design Partners and Site

This co-design project brought together a multi-university and cross-disciplinary team of researchers that included expertise in social sciences, medieval culture, design methods, human-computer interaction, and community-led innovation. All research activities were held at St Nicholas' Church Brockley and were supported by The Friends of Brockley Church and The Churches Conservation Trust (CCT).

St Nicholas' Church (an Historic England grade II* listed building) developed from a small Norman building largely in the 12th and 13th centuries (Figure 1). Most of its furnishings, except for its exceptional pulpit and font, come from a restoration of the early 19th century [33]. St Nicholas' is cared for by the CCT, a statutory body and national charity caring for closed churches transferred to it by the Church Commissioners (who manage the property assets of the Church of England). Although still consecrated, it is no longer in regular use by a congregation, that role being served by a second historic church in the same parish. Brockley, a small village and civil parish (of less than 700 acres) with a population close to 250, lies close to a primary road connecting two important regional

centres (Bristol and Weston-Super-Mare). Historically a manorial estate (without a 'village centre'), modern Brockley consists of small, independent clusters of residential buildings that have emerged from, or around, estate property. There is currently no pub, post-office, school or community hall (this was also historically the case). Whilst farming dominated community life in the past, the local economy is now a diverse, modern one, with 19% of residents being small employers or sole traders (in the last available census) [34].

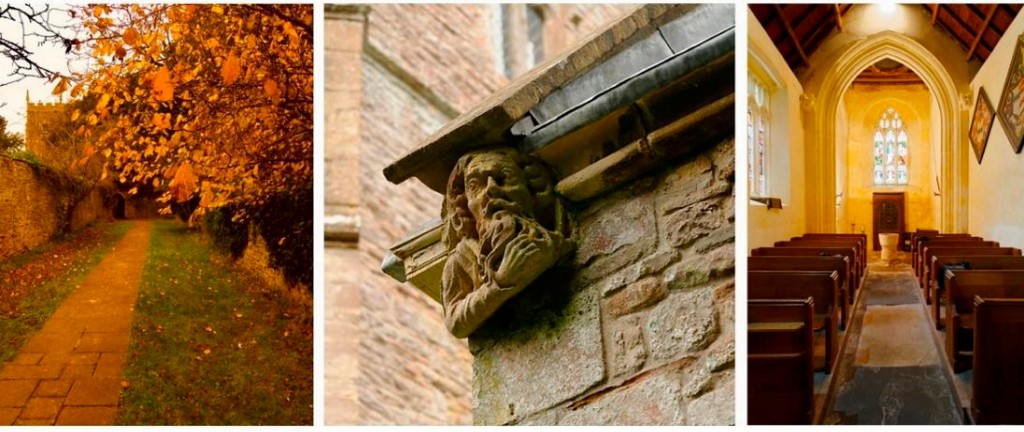

**Figure 1.** St Nicholas' Church, Brockley.

### 3.2. Theory of Socio-Cultural Viability

Clumsy solutions consider a plurality of perspectives and rationales to find more effective shared solutions with the broadest appeal to a common (Wicked) problem. They call for action in the social domain. An important theoretical grounding for their development has emerged in the theory of socio-cultural viability, Cultural Theory for short [35–37]. Cultural Theory argues that social life can be described in terms of interactions between four fundamental ways of organising, justifying and perceiving social relations: hierarchy, egalitarianism, individualism, and fatalism. It argues that the ever-changing mix of these four building blocks (or ways of life) generate the endless diversity we see in the social domain.

Underlying these four building blocks are two dimensions of sociality: 'grid' (status differentiation from high to low) and 'group' (collectivity from high to low) (see Figure 2). In this way, Egalitarianism (E—high Group, low Grid) is defined by high levels of group membership (whether through shared resources, residence, work or recreation) and strong group boundaries with minimal role prescriptions and role definitions; Hierarchy (H—high Group, high Grid) characterises individuals whose social environment has strong group boundaries, binding prescriptions and clearly designated roles, giving individuals restricted autonomy within that environment; Individualism (I—low Group, low Grid) describes individuals who are bound neither by group incorporation nor prescribed roles (with all boundaries provisional and subject to negotiation), experiencing high levels of choice in how they spend their time, with whom they associate, where they live and work etc.; finally Fatalism (F—low Group, high Grid) characterises those who find themselves subject to binding prescriptions, are excluded from group membership, are controlled from without and enjoy little individual autonomy (see Figure 2).

These four ways of organising and perceiving social relations both oppose and are dependent on each other, achieving stability through interaction. To paraphrase Schwartz [39]: Individualism would mean chaos without hierarchical authority and shared solidarities. Hierarchies are stagnant without the creative energy of individualism, incohesive without the binding force of equality, and unstable without the passivity and acquiescence of fatalism [39]. It is these fragile alliances that can produce resilient, adaptive, and flexible social life. Whilst forms of social action (e.g., governance) that impose only a single way of organising, perceiving and justifying social relations is vulnerable to failure [40], the

search for clumsiness, in contrast, asks how fundamental social building blocks might be creatively and flexibly *combined* to generate more resilient, shared solutions. The empirical evidence on the utility of Clumsy solutions is stacking up [41].

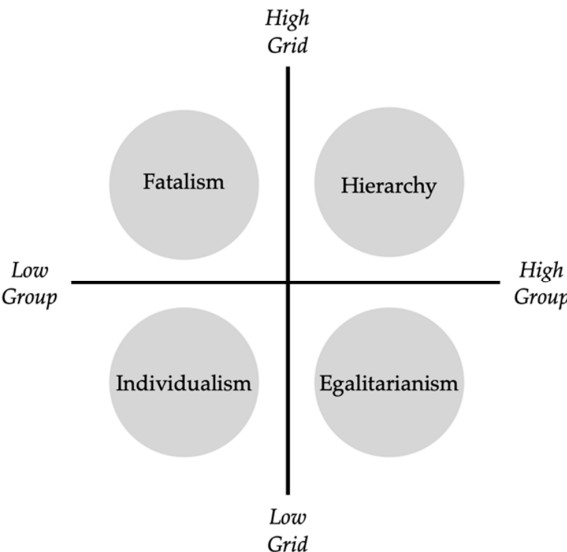

**Figure 2.** The four fundamental ways of organising, justifying and perceiving social relations according to Cultural Theory, adapted from [38].

*3.3. The Search for Clumsiness*

In the search for Clumsy solutions, we have deployed Cultural Theory in a three-stage process: Stage 1—Defining a trajectory for clumsiness from a community profile; Stage 2—Workshop design for clumsiness; Stage 3—Finding drivers for clumsiness in long history and slow technology dimensions. This process is summarised in Table 1 and Figure 3.

**Table 1.** A Three-Stage Process for developing Clumsy Solutions.

| Stage 1 | Stage 2 | Stage 3 |
|---|---|---|
| Defining a Trajectory for Clumsiness (Section 4.1) | Workshop Design for Clumsiness (Section 4.2) | Finding Clumsiness in Long History and Slow Technology (Sections 4.3 and 4.4) |

**Stage 1**—Defining a Trajectory for Clumsiness: In this first stage, we worked with Cultural Theory to generate a snapshot of how communities facing uncertainty, and an unused church, construct and understand their own social realities. An initial round of nine qualitative interviews with ten individuals (20–60 min each) was conducted between January and March 2020, adopting a snowball sampling method initiated through the principal CCT volunteer at the site. Using a walking interviews methodology, interviews focused on issues of community life today (e.g., health and wellbeing, and social connectedness) and perceptions of St Nicholas's church. Qualitative data was analysed using a constructivist grounded thematic approach [42] to identify themes and patterns corresponding to the four social building blocks of Cultural Theory: a community profile. From that profile, we defined a desirable 'clumsy' state—the creative re-combination of those self-same building blocks to generate an alternative form of social life that may benefit the community and its church—and a 'clumsy trajectory' towards achieving that state. This work is detailed in Section 4.1.

**Stage 2**—Workshop Design for Clumsiness: Our second stage was to develop co-design workshops that can begin the transition towards the desired clumsy state, i.e., to design workshop structures and activities that work with the community as it is now (as described in the profile) whilst also moving them along the clumsy trajectory defined

towards a different (clumsier) understanding of church/community futures and their role in it. As each of the four ways of life have their own way of generating solutions (from how participants are recruited for workshops to how discussions are conducted and decisions made [41]), we once again turned to Cultural Theory to design workshop structure and content. Workshops followed on from the interviews with a 19-month lag, in adherence to the COVID-19 social distancing guidance of all project partners. (Informal conversations within the community prior to project restart indicated that perspectives on the church and the community were broadly unchanged, reflecting long-standing concerns). Seven workshops were delivered at St Nicholas' church through November and December 2021. These were attended by between six and ten participants per event (19 individuals overall). In this case, 20 future activities making use of the church in the community were developed over the course of the workshops. This work is detailed in Section 4.2.

**Stage 3**—Finding Clumsiness in Long-history and Slow technology: The central purpose of the workshops was to explore how ideas from the long history of parish churches and the affordances of slow technology might bring the desired clumsy state into being. Long history themes were initially selected for their fit with thematic patterns identified in the interview material. These themes were further developed through establishing background historical contexts and some original archival research. Themes that captured our target clumsiness were pinpointed during workshops, and their practical implications developed over the course of the programme. This work is detailed in Section 4.3. In parallel, we introduced a slow technology approach into workshops built on the JigsAudio system [29]. JigsAudio uses Radio Frequency Identification (RFID) technology to associate audio files with physical objects. It was originally designed to help encourage people engage with topics in a slower, more reflective manner through making and playing-back audio recordings associated with tangible (physical) tokens. The device consists of a Raspberry Pi, an RFID tag scanner, a battery and a microphone. To use the device, participants place a token on the device and press a record button to begin and end their recording, which will associate that audio recording with their token via its RFID tag. Later, participants can listen to their recording by placing their token on the device and pressing the 'play' button. To-date, JigAudio has helped encourage participation in projects on the future of the Tyne and Wear Metro and the use of green spaces in Newcastle, amongst others (http://jigsaudio.com/ (accessed on 30 March 2022)). JigsAudio was introduced into workshops in two different forms: the original and a modified version designed to resonate with the ritual and material heritage of the church site itself. These two versions were introduced as provocative prototypes (called "provotypes" [43]), i.e., as examples of how slow technology might be used in conversations about church futures, a way to stimulate discussion on slow technology's particular (and potentially clumsy) affordances in that role. This work is detailed in Section 4.4.

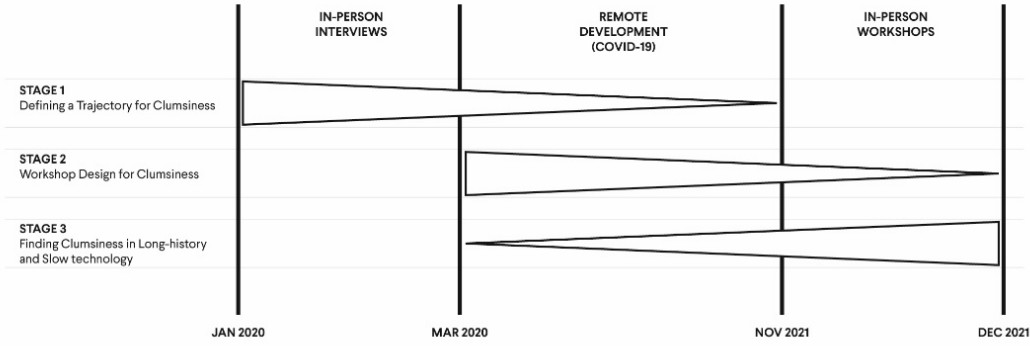

**Figure 3.** The three-stage process for developing clumsy solutions as applied at Brockley.

## 4. Co-Designing Church and Community Futures

### 4.1. Defining a Trajectory for Clumsiness (Stage 1)

4.1.1. A Cultural Theory Analysis

In this first stage, we worked with Cultural Theory to generate a snapshot of how a community facing uncertainty, and an under-used church, constructs and understands its own social reality, and how these constructions are embedded and constrained by local institutional cultures. This analysis broadly outlines how four fundamental ways of organising, justifying and perceiving social relations (hierarchy, egalitarianism, individualism, and fatalism) present in the community.

**Egalitarianism:** Brockley's identity as a community, and even a village in a strictly geographical sense has been questioned by residents. Rather than a 'single community', plural forms (communities, friendship groups, circles) were regularly used, each group describing a different experience of village life and intensity of social connection. In the words of one resident: "there certainly are communities, I'm not sure that really at the moment there is a single feeling of community for Brockley . . . it is as a result I suspect of not having any form of, sort of, you know, collecting point" (Interviewee 1). This was echoed by another interviewee: "It's an overall perception that in this tiny geographical area, perhaps, we could all know each other, but we don't. And I think some of that is [caused by] a physical boundary . . . There's no central point of cohesion, if I was to try and put it in a sentence, and I think that feeds into this 'separate group' thing" (Interviewee 2). The sense of intra-group support can be very strong, in part due to the many long-term residents who have been living in the village over decades, for example: "you can literally draw a circle round it, that is, who I see as my immediate neighbours . . . and they are phenomenal and anybody in that [circle] could knock on their door and say I'm unhappy or I've got a problem or I've got a crisis, and they would do something." (Interviewee 3). There is, none-the-less, a sense of neighbourliness within the parish, for example: "I guess there's not so much intermingling as you might like or expect, I don't know. But you know it's not to say it's unfriendly or anything. It's quite a friendly place . . . you know everyone knows everyone else at least from sight or first name terms" (Interviewee 4). Against this background, the need to increase social interaction across the village, however informal or casual, was considered highly desirable, but also increasingly hard to achieve, whether through a lack of places to meet, residents living their lives beyond the village, or the increasingly digital nature of everyday interactions. As one resident described: "the gathering together of people in places is good for the soul, and we need that. Communities offer that if you can recreate them, because I think we are beginning to lose them . . . There just needs a rethink, I think, about not just what's on offer [in the village] but how you create a community, how you draw those threads together" (Interviewee 2). In the absence of traditional community sites for gathering together (with no post-office, pub or village green), social life takes place by-and-large in private settings, a scenario that can, for some, unintentionally foreground differences in perceived wealth and social status within the parish.

**Individualism**: In mirror-image, residents described themselves (and others) as living largely independent and self-sufficient lives. With less attention focused on community life beyond established social groups, interaction may only take place when there is direct reason for it. As one resident put it: "people aren't always looking into the centre, they carry on their lives independently of Brockley" (Interviewee 1)". The increasing number of 'gated' houses in the village are also seen as closing-off social and serendipitous encounters that might otherwise encourage a sense of community. Several residents described the importance (for themselves and others) of looking outwards beyond the village to build relationships that have deeper meaning in terms of their individual working lives, leisure interests and family needs etc . . . This is in no small part facilitated by Brockley's ease of connection to important regional centres: "people enjoy it for its partial isolation but at the same time you can get anywhere from it and its well-connected and Brockley in a way is better connected than it's ever been in its time to, you know, the region [and] the

capital" (Interviewee 4). As a result, it can be a challenge for those wishing to mobilise social connection or social inclusion across the village, with some reluctance to think in new ways about the nature of community life. For some, that driver of social change is only going to come from individuals who want it: "We do activities during the course of the year, a bit samey, but . . . we should be ready to change and that's where I think if I want something to change I've got to get up and do something about it" (Interviewee 3). In a broader sense, a strong measure of individualism within village life can be found in the highly diversified nature of local businesses and enterprises.

**Hierarchy**: Brockley parish has many traditional governance and decision-making structures in place. Notwithstanding the Parish Council, many of those structures concern St Nicholas' church. These include the Parochial Church Council (the executive committee for the parish), Church Friends group, Churches Conservation Trust, and the equivalent committees of the active historic Parish church nearby. Participation in committee activities and public consultations within the village is low, and there is often little change-over in committee membership (sometimes across decades). Whilst, for some, the unchanging nature of village life is a virtue (a sign of stability), for those seeking change, it acts as a source of resistance. As one resident put it: "it is like a contagion in rural communities and not necessarily even rural communities, it is this: this is how we do things, don't rock the boat, don't upset people, don't question things, this is how it's done—there's a lot of that" (Interviewee 2). The combination of 'church and committee' is recognised as a particular challenge: whilst a number of important successes have been delivered for the church (including maintenance and new facilities), the multiple committee structures involved can mean decision-making is slow, time-consuming and protocol-driven (with membership often seen as an obligation). Since traditional committee structures tend towards a hierarchical and 'low-trust' model, they can favour accountability (and maintaining the status quo) over innovation. Further, they may be procedurally unable to respond to a world different from that for which they were created. In the end, driving change within an organisation falls to the persistence and hard work of individuals, even when that change is deemed necessary and has broad consensus. With little appetite locally to join committees, there is space for new models of participation and decision-making.

**Fatalism**: At its core, fatalism (a 'whatever will be, will be' attitude) describes a sense of having little control over key events happening around you: one is, in essence, subject to events driven by others and from elsewhere. In our conversations about the future of the community, a number of barriers to a re-invigorated community life emerge, many of these seen as being outside of reasonable influence: The first might be described as a struggle to reach a self-sustaining 'critical mass' of social activity, compounded by a low participation in local organisational structures and few ready to lead on change; the second concerns the lack of a suitable venue for social gathering (one always measured against the characteristics of a traditional village hall or pub) and the fragmented geography of the village itself; third, the triumvirate of an ageing demographic, the near-absence of a younger generation (recognised by many in the village as a critical driver of community identity) and the lack of affordable housing that could sustain a 'connective' middle generation; finally (and in direct tension with the third) is an external force in the form of proposed housing developments in the immediate locality, seen as an inevitable response to housing pressures but one with consequences for the community as a dormitory village on a busy commuter route. Resisting fatalism is a recognition that a reinvigoration of social life—community life as a driver of change from within, not just subject to change from without—is possible but will require active intervention. As one interviewee put it: "I think it [will come from] a group of people who are prepared to push the boundaries a bit and be prepared to put some energy into it—to make change . . . I get a sense (and I probably include myself in this) that people will talk about being willing to have change but somebody has got to be prepared to go out there and make it happen. And if people lead busy lives or have other commitments it's very hard to make things happen. That's the challenge, because you are sort of pushing against something that hasn't changed in a long time, and I do

get a sense there's a bit of resistance. I don't think its active resistance, it's just passivity." (Interviewee 3).

4.1.2. A Clumsy Trajectory

Four broad-brushstroke characteristics emerge from this profile of community life: a diminished sense of social solidarity (a sense of belonging to a community with a shared purpose); high levels of individualism at the expense of shared interests locally; a loss of engagement with traditional structures organising community life; and a degree of fatalism towards this situation improving in the future. The same analysis, however, reveals an alternative side to the profile: a weaker sense of 'community in place' reflects richer 'communities of interest' beyond the village that might be brought to bear on village life; high levels of individualism reveals a more diverse source of interests and life experiences that could enrich community interactions, and; limited engagement with parish, parochial and local forms of governance may suggest the need—and appetite—for new types of participation, representation and decision-making locally.

St Nicholas' holds an important position in this profile, one mirroring Brockley at large. Although the only public building in the village, the church is rarely in use by the community, holding 2–3 services and a handful of one-off events each year (although historically more so). The church has a small but committed volunteer base (Friends Group) who provide basic site management, support 'visitor welcome' (estimated at around 100 visitors per month) and play an important role in fund-raising. A number of entrepreneurial individuals are seeking new developments for the church, a process that could be better supported by current organisational culture (in and around church decision-making). As a historic church, interventions into the church fabric (for example the removal of pews) are highly restricted, limiting for some (without redress) future use of the building as a community venue. With an ageing demographic, there is concern around sustaining an active volunteer base and responding to rising maintenance and repair costs (which are not currently being met locally).

Yet, for all interviewed, the church is a special place that holds many different dimensions of meaning: valued for its art and heritage, as a site of peace and spiritual refuge, and as holding significant value for both families (e.g., through christenings, marriages and burials) and the community at large. As one interviewee described: "I think of the church as, in a way, the centre of the community because it's the thing that links everybody together . . . in a way, potentially, it does link everybody together. I think of it more as a community place rather than a religious place" (Interviewee 5). In the view of another resident, there is a clear vision for how the church might serve the community in a broader sense: "I want this place to be an exemplar of what you can do creatively in a space, in a small community that provides a purpose, not just to the building but provides somewhere [for people] to go . . . have a glass of wine (or a non-alcoholic drink), and know that there would be maybe 4 or 5 (maybe 20!) other people here, and once a month sit down and have supper with them, and have a local music group, and a . . . I'd like it to be a beacon" (Interviewee 2). As such, St Nicholas' holds considerable potential as a site of 'clumsiness'—a way to think differently about the social future of St Nicholas and Brockley together. Working from this community profile, a clumsy trajectory for the church and community readily suggests itself, consisting of four aims:

T.1.  Redirecting individualism towards social asset building through the church (from Individualism to Egalitarianism, **I→E**);

T.2.  Enabling a more pluralistic account of community and church 'ownership' (increasing Egalitarianism, **↑E**);

T.3.  Offering alternative structures for representation and decision-making in church use (reducing or challenging Hierarchy, **↓H**);

T.4.  Bolster/demonstrate new types of agency in church futures (tackling an undercurrent of Fatalism, **↓F**).

*4.2. Workshop Design for Clumsiness (Stage 2)*

4.2.1. A Clumsy Workshop Structure

Co-design workshops were developed directly in response to this clumsy trajectory, putting measures in place to encourage active participation from those with little current connection to the church, strengthen a sense of solidarity in working together on church futures and offer a more open and informal collaborative process (a flatter hierarchy).

Responding to the individualism dimension of our clumsy trajectory (T1), workshops were advertised through as many different channels as possible (from word-of-mouth to residents' associations and newsletters). Workshops were kept short (around an hour), were open-ended in terms of commitment (with the opportunity to participate kept open for the duration of the programme), and held weekly on alternate mornings and evenings to widen access for different age groups. During workshops, participants were encouraged to introduce new, bold, or unconventional ideas (both in-person and through the JigsAudio slow technology device) and to actively reach out, from week to week, to others who might wish to participate.

From the perspective of strengthening solidarity (T2), workshops adopted a relaxed peer-to-peer mode of engagement supported by collaborative notetaking, mapping, and brainstorming. These activities were carried out in St Nicholas' church, using spaces suited for close group working (including a unique "parlour" pew). Whilst individual workshops were carefully themed (see below), emerging shared interests shaped the direction and sequencing of workshops. Workshop conversations sought to identify, through widespread agreement, clumsy solution drivers with broad appeal.

With respect to reducing perceived hierarchy (T3), a light-touch approach was adopted in running workshops, creating distance from conventional committee structures. This included (as described above) open recruitment, adaptive programming, collective note taking, and the self-assignment of tasks (e.g., in idea development). Whilst each workshop focused on a broad thematic area, incorporating planned activities and introducing expert insight, the sequencing of workshops and activities was made highly responsive to emerging ideas. The analysis, synthesis and summarising of workshop insight helped bring concepts forward through the programme for reappraisal.

Finally, fatalism, too, had an important role to play in workshop dynamics. In keeping workshop attendance always open, workshops could break free from a higher-level predictability, with conversations often taking unexpected turns in response to new participants and changing participant dynamics. In workshop conversations, recognising uncertainty in finding a solution or promoting empathic engagement with those who might reject your ideas (both eremitical tools) offer ways for participants (and facilitators) to question their own biases and expectations. (Eremitical tools draw on the idea of the hermit, the fifth way of life distinguished in Cultural Theory, as a mode of becoming temporarily distanced from one's own social contexts and cultural biases when making decisions). With workshops developed from week-to-week in response to emerging conversations, the exact clumsy solution that might emerge could not be known in advance. In this way, conversations might better reflect 'where people are' over the course of an extended engagement. This all serves to make room for unexpected voices or for new voices to have potential impact (T4).

4.2.2. Workshop Content and Activities

Putting this into action, the workshop series began (workshop one; 8 participants) by exploring current conceptions of the role of church buildings, using a simple "what 3 words?" activity to capture what St Nicholas' means to people. Key historical concepts about the multi-functional nature of church buildings and the legitimacy of tapping into such histories were explored (see Section 4.3). The "what 3 words" activity was revisited at the end of the workshop to think to the future of the church people would prefer to see (see Table 2. The second workshop (8 participants) started to add substance to that 'future', using a collective mapping exercise to document the histories, heritage, folklore, life events,

people, organisations, businesses and geographies that animate parish life today. A model of the connectedness of medieval life was used to re-situate church, community and parish (an overlaying of different mental topographies) 'in the same place'.

**Table 2.** Responses to the "what 3 words?" activity in workshop 1.

| What Three Words Describe How the Place Makes You Feel (Selection) | What Three Words Describe How You Want the Place to Be (Selection) |
|---|---|
| Cold, Sad, Lonely | Warmth, Vibrancy, Community |
| Unique, History, 'Hidden Gem' | People, Life, More activities |
| Tranquillity, Beauty, Age | Conversation, Life, Warmth |
| "Long historical ties" | "Maintain that history" |
| Compact, Attractive, Calm | Populated, Busy, Colourful |
| History, Beauty, Stained glass | "Keep the mystery" |

Prompted to explore how participants might reconnect with their church, workshop three (8 participants) focused on St Nicholas, the church's patronal saint: including legendary biography, liturgical celebrations and festivities, apocryphal stories, and the dynamic and shifting nature of Saintly patronage across the Middle Ages. A simple game was devised to explore (guess, promote, refute) possible patronage attributions. Here, an exploration of St Nicholas' gift-giving proved an important step in thinking about a future placemaking strategy. Having established the value of the Gift in workshop three, workshop four (8 participants) returned to the earlier mapping activity, extending it beyond the parish to include regional connectivity and creative/cultural influences—a way to uncover more fully interests people's interests and potential offer. Here, participants' interests in social prescribing (community referral for wellbeing), local, social and natural histories, regional 'creative ecologies' and pilgrimage routes (past and future), amongst others, were discussed (Figure 4).

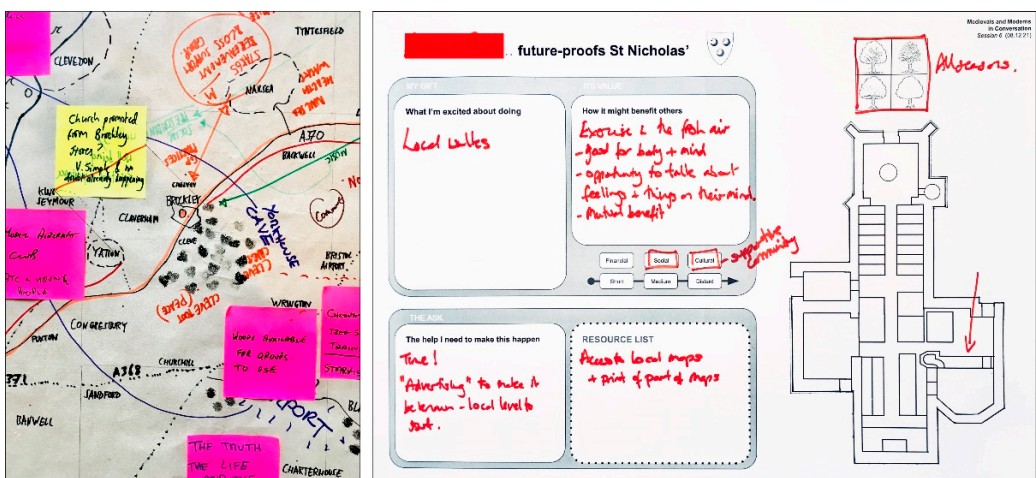

**Figure 4.** Workshop materials: parish mapping (**left**) and Future Proofing canvas (**right**).

This renewed mapping activity prompted, in workshop five (7 participants), a closer examination of the church's current messaging: how signage, access, and internal/external environments speak to the role and values of St Nicholas' today. Vision, Mission and Strategy templates were used to propose alternative futures, incorporating key ideas from earlier workshops about 'what matters' in a sustainable future for the church. The annual patronal service for St Nicholas took place between workshops five and six (the one regularly scheduled event each year). The service provided an opportunity to introduce the project to the wider community and recruit for a special event where our emerging model of 'gifting and redeeming value' could be further tested.

That event (workshop six; 10 participants) worked with a bespoke 'future proofing canvas' (Figure 4) where participants (in pairs) were invited to put forward an activity (their Gift) for the community, considering gifting and value trajectories, resources and activity support, timeframe (a seasonal calendar), and wider local/regional contexts. A demonstration of Crumhorns (a double-reed capped wind instrument popular in the 15th and 16th centuries) organised by the lead author was used to explore the affordances of different church spaces for music-making as one of the future proofing projects. The final workshop (workshop seven; 6 participants) reflected on the creators of, and audiences for, these different projects designed to 'future-proof' the church. Here we sketched out different personas, mapping barriers/enablers to participation.

Slow technology prototypes were used throughout the workshop programme to ask new questions of slow technology as well as drive, inspire and document participation (see Section 4.4).

*4.3. Finding Clumsiness in Long History (Stage 3)*

Working from the initial thematic analysis of interview material, our long history research sought to identify site-specific and broader histories (e.g., of social movements) that reveal the changing and adaptive nature of rural parish churches in their communities. These included the different roles a church building and its clergy would serve in the community historically and across the life course; the close interweaving of commercial, religious and secular lives; tradition, dissent and 'acting out' in the Church; the role of social capital and mobility within/beyond parishes; gifting and exchange, amongst others. Introducing these long history ideas into the workshops (both through focused conversation or tangentially through activities), five formulations held particular resonance, each suggesting how St. Nicholas' long history could serve the clumsy trajectory introduced earlier (see Table 3 for summary).

**Table 3.** Summary of the long history dimension and how it enables clumsiness.

|  | T1: (I→E) | T2: (↑E) | T3: (↓H) | T4: (↓F) |
|---|---|---|---|---|
| **Multiple Church Roles** | Increases the range of potential uses with community in mind | Invigorates a pluralist account of churches as social networks | Offers a counter-argument to siloed thinking on church use | Recognises the importance of diverse voices |
| **One Space for Everyone** | Defines shared space that resonates with people differently | Reveals a variety of spaces for communal experience | Opens up authoritative spaces to a broader range of people | Opens up spaces that can be made to "speak to someone" |
| **Looking Inwards and Outwards** | diversifies resources for building shared assets | Offers a richer and multi-layered account of community life | Disrupts siloed working and thinking locally | Recognises other sources of influence that could matter |
| **Gifting and Redeeming Value** | Broadens what is of value to the church and community | Helps build historical, present and future solidarities | Empowers grassroots overtop-down recruitment. | Invites participation to show what's possible |
| **Circularity in Time, Nature and Economies** | Takes pressure off people with busy schedules | Affirms the value of sustained collective action | Challenges a default to committee-led decision-making | "Restarts" time by giving it meaning once more |

4.3.1. Church as Multi-Role

The long history of medieval rural churches reminds us just how integral they were to everyday life [44]. Whilst Christian teaching would have been central to their work, they would also have been a social service, a meeting place, law courts (for enforcing canon law, which affected many everyday activities) and a driving force behind creative, artistic and even commercial activities (in the churchyard if not the church itself) [45]. Through the organisation of the parish church, a community could raise revenue from lands and bequests to support church building and the 'divine services', but also support

local infrastructure, provide credit, poor relief, and basic education. These roles would have woven communities, individuals, powerful families, and guilds into a complex and dynamic parish life. In this way, much of what we consider today as subdivided into distinct religious, secular, and commercial pursuits would have been bound together through the Church and its parish in response to a community and its needs (for example in the role of the vestry committee, itself a micro-laboratory for models of community organisation; [46]). Further, the parish church would have been a place of performance (whether part of regular liturgical worship or processions) that would stand in opposition to contemporary distinctions between sacred and profane [47,48]. The question of what 'non-worshipping' activities are appropriate in a place of worship is neither simple nor trivial. At a time of crisis, where people are looking with urgency to bring new uses to churches, recognising this deeper historical clumsiness can serve as a counterpoint to arguments of 'one church, one purpose'.

From the perspective of Cultural Theory, this multi-role characteristic expands the historical precedent for how churches have been used, by whom and for what community-oriented purpose (T1: I→E); it reinforces a pluralist account of community, foregrounding the church as a 'social network' responsive to different potentials and needs rather than as a 'homogenising' and centralising hub (either for "the community" at large or for those of a Christian faith in our modern, more narrowly conceived sense of the term) (T2: ↑E); it defuses an institutional (and self-policing) model of 'appropriate' church-use in two ways, first by recognising how cultures of use are only ever contemporary (and, therefore, subject to change historically and in the future), and, second, that other forms of social influence have long played a role in shaping church use (T3: ↓H); it demonstrates not only historical precedent for doing things differently but affirms different types of agency in sustaining the life of a parish church (T4: ↓F).

4.3.2. One Space for Everyone

Historically, rural churches were the only communal space open to everyone and to which anyone could participate/contribute over their whole life course. As such, church buildings themselves became microcosms of the societies and social structures that built them. Medieval life conceived of many forms of familial, friendship and social structure that bound people together around shared interests and invigorated the material nature of the church and the cycle of services and festivities of the church calendar [49,50]: furnishings, artworks, artefacts (temporary and permanent) would all leave an accumulative mark. The 'space' of the church was, therefore, more than just 'a physical container' but multiplicitous: a site, a route, a setting, a refuge, a pathway, a showcase—a place bridging temporal and atemporal worlds. Today, the question of how to prepare churches for greater community use is a controversial topic, with the removal of Victorian and Modern pews a popular remedy to expand the availability of *all-purpose* space within a building (resonating with Egalitarian principles). The long history of church sites, however, points to the inherent clumsiness of these multiplicitous spaces as a source of potential rather than a problem.

From the perspective of Cultural Theory the historic parish church can offer a variety of spaces for quiet reflection, refuge, inspiration and creativity, with an accumulated, shared heritage that will speak to different people in different ways (T1: I→E); public space also abounds—suitable for interaction, exchange and co-production; consider box pews with internally facing seats or the area around a font (the 'universal equaliser'). That shared space can extend beyond the church itself, blending with public services (e.g., transport) and connecting into the community (e.g., in the use of private amenities, such as toilets or kitchens, for time-limited public use) (T2: ↑E); the intense differentiation of spaces along hierarchical lines (e.g., pulpit, pews along a church axis, chantry chapel or family/parlour pew) all lends themselves to activities with an element of authority, but with mutually beneficial exchange in mind—whether that's giving a lecture or presentation, directing a workshop, or offering guidance (T3: ↓H); this potential to create spaces that "speak to

someone" (neutral or charged) may help engage with people who see church futures as outside of their interests and influence (T4: ↓F).

### 4.3.3. Looking Inwards and Outwards

Building on the multi-functional and multi-spatial nature of the medieval rural parish church, its influence would have extended beyond a narrow focus on 'village life': it would foreground an understanding of family, kinship relations, community, parish identity, interactions with other parishes (through trade, apprenticeships, festivities, and rivalry), encompass national and pan-European influences, and, finally, invoke an all-encompassing common heritage through the Universal Church. Socially, for example, the medieval parish becomes instrumental in embedding "individuals in a new horizontal framework emerging alongside kinship ties and vertical bonds of subordination, with access to offices and public responsibilities" [5], paving the way towards a more connected and post-feudal society. The perceived isolation of rural and rural-urban settlements belies the enormous connective capital of individuals and communities today. We argue this long history dimension can invigorate a reappraisal of 'what connection looks like' (in terms of geography, social movements, professional lives, ideas and influences etc . . . ) and ask how the story of the parish church might once again foreground that connectivity in a clumsy way as, perhaps, it once did.

From the perspective of Cultural Theory, someone's social capital within and beyond the parish may be a source of inspiration (or strength) for the community. Here, particular emphasis is placed on the variety and extent of those connections, activities that tap into new sources of ideas (e.g., the regular stream of CCT visitors from outside of the parish), and the entrepreneurial thinking that can draw a thread between potential and real-world impacts (T1: I→E); increasing the variety and sources of influence in parish life—looking both inwards and outwards—is not only consistent with the pluralist nature of communities today, but may benefit them for the same reasons: driving new, and perhaps unexpected, forms of connection and friendship (T2: ↑E); a renewed emphasis on what connects people within and beyond the parish was identified in workshops as a means of disrupting long-established, siloed working amongst local bodies, community groups and local businesses (T3: ↓H); this diversification may empower different types of community actor because the sources of influence that come to matter may differ from those of the past (a broadening, not a narrowing) (T4: ↓F).

### 4.3.4. Rethinking St Nicholas

Perhaps the most significant of the themes is a re-imagining of patronage under St Nicholas in a way that foregrounds the multi-role, multi-site and multi-capital potential of the church in its parish today. As Nicholas the Wonderworker, Saint Nicholas (an early Christian bishop of Greek descent) is the 'heavenly advocate' for many dozens of groups: the Patron Saint of children, coopers, fishermen, merchants, broadcasters, the falsely accused, repentant thieves, brewers, pharmacists, archers, prostitutes, pawnbrokers and unmarried people, amongst many [51]. This enormous variety comes from an interaction between local and ecclesiastical concerns unique to each place at a given time, generating "the idea of a Patron Saint" that speaks to—and inspires—a whole community. The formation of local guilds to support festivities, church refurbishment and fund-raising would promote that unique sense of connection all year long [47]. The underlying concept of his patronage is one of Gifting and Redeeming Value, where the notion of the gift, its associated value and the time-course over which value is redeemed are all open to considerable interpretation. The long history of St Nicholas is a stark reminder of how dynamic and historically contingent patronage is, one able to find a new voice in response to new demands. By looking beyond a narrow formulation of present-giving associated with the saint, considerable clumsy potential emerges.

From the perspective of Cultural Theory, everyone has the potential to offer something in the context of church futures, whether, that's offering or facilitating an activity, offering

organisational support, or providing a specific skill such as social media support. By broadening what is of value to the church and community, the things people enjoy doing (their gift—willingly and joyously given) is more likely to find meaningful resonance with others (T1: I→E); Under the auspices of St Nicholas, this becomes both a community-enabling concept relevant to today and a form of historical solidarity, one able to support a variety of gifting-valuing transactions (cultural, social, financial etc.) (T2: ↑E); it reconfigures the relationship between the community and Church groups, ostensibly from "what can you do for the church" or "we need you to do x, y, z" to "what do you love doing, that you want to share with others, and how can this church help you?" This top-down to bottom-up inversion broadens-out conversations on the nature of value (what is valuable for church and community) and the alternative organisational forms that can help realise it (T3: ↓H); by inviting others to make an offer of a gift in the best way they can, the opportunity for meaningful participation and impact (the redeeming of value) is increased (T4: ↓F).

### 4.3.5. Circularity—Time, Life, Nature, Economies

Historically, real-life experience would be an important marker of time, shaped by annual cyclical patterns of worship, festivities, and the seasons (planting and harvesting) [16,50]. The festival of St Nicholas (6th December), marking the end of autumn and the threshold to winter, was celebrated by young people roaming about in search of food or money [16], both embracing and defying the lengthening nights: "By the time of his feast day, Advent Sunday had brought once more the promise of salvation and victory over the powers of darkness, as well as the prospect of the great midwinter feasting-period. And so one cycle ended and the next began" [52]. The cyclical nature of time found important resonance in an understanding of the lifecycle, cycles of gifting (to sustain, renew and connect generations), and circularity in production (reuse, repair and repurpose), and so on. By contrast, the construction of time is a problem for modernity: in the current debate, the pressure of 'linear time' is strongly felt, marked by an ageing demographic, generational loss as opposed to renewal, the increasing pressures of church repair and maintenance, notions of progress, and so on: the crisis of historic churches and rural communities is seen as contemporary and marching in one direction. This finds parallel in approaches to church planning [e.g., 25], where crisis response is measured by a linear, unidirectional movement from diagnosis to solution and "managing the new normal".

From the perspective of Cultural Theory, a shift of emphasis in church programming from open-ended to annual (pivoting around the annual patronal day) reduces the pressure on those for whom time is a precious and limited resource ("would you like to do something this year?") (T1: I→E); it affirms the significance of action by recognising a finite period in which a collective commitment can be made, setting the foundation for the next annual cycle of activities that sustain community life (T2: ↑E); it can challenge the easy default to regular, formal programme meetings under the pressure of "now, what next?" (and the committees that go with them), introducing instead a slower cycle of development, testing, reflection, iteration and discourse (T3: ↓H); it can "restart" time again for the fatalist, fostering an awareness of distinct seasons and their potentials by retuning into the world around you (T4: ↓F).

### 4.4. Finding Clumsiness in Slow Technology (Stage 3)

#### 4.4.1. JigsAudio Provotypes

It was through JigsAudio [29] (open-source technology that supports people to engage with topics through associating tangible 'tokens' with digital audio recordings) that we explored how slow technology might serve a clumsy placemaking strategy. In contrast to technologies that allow people to quickly give their views, JigsAudio is designed to encourage reflection and dialogue. It does this in two ways, first by encouraging people to reflect on a topic before committing to making a recording (a prompt to share more thoughtful, closely held ideas, aspirations or provocations for the space), and second by

giving people the freedom and permission to express their ideas creatively and in their own voice (whether as an individual or a group).

Two versions of the JigsAudio device (Figure 5) were used between workshops to help participants reflect on our conversations and introduce additional ideas into subsequent sessions. They two versions served as provotypes (provocative prototypes [43,53]), i.e., as examples of how slow technology might be used in conversations about church futures, simultaneously supporting interaction and prompting further discussion on the affordances of slow technology in placemaking. Our second version was developed during the COVID-19 pandemic (in the 18 months between the stage 1 interviews and stage 2 workshops). Here, modifications were made to help focus interaction on device-use (rather than the device itself) and respond to the specific ritual and material affordances of the church site. By hiding the hardware from view through new casing, simplifying the button interaction (play and record buttons replaced by a touchless token reader), and using in-built speakers (so eliminating the need for headphones), the device becomes less of a 'techno-scientific object to be understood' and introduces a more seamless moment of interaction that helps the user focus on what they want to say, not how to record it. The new casing was designed to echo the heritage of the church site, adopting a bowl-like quality that draws on the symbolism of the ciborium (a sacred vessel with important symbolic and communal dimensions) without imitating a religious object too closely. As a bowl—an invitation to be held—the device is made more mobile for use anywhere in the church. The inner, stepped surface of the device depicts a model of the medieval universe as imagined by Dante Alighieri at the beginning of the 14th century in the Commedia. A love poem, the Commedia tackles the pursuit of a good life in the face of lived uncertainty, the expression (to paraphrase Robin Kirkpatrick) of an appetite, shared with all other forms of life, to live as completely as possible [54].

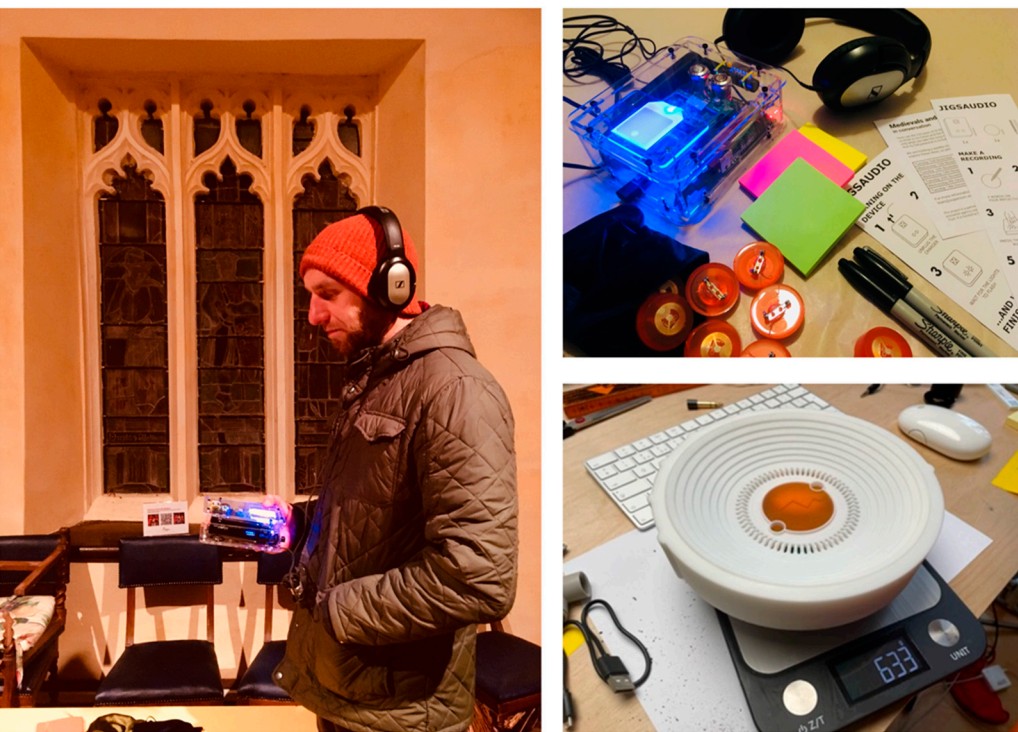

**Figure 5.** Two JigsAudio provotypes. The original (**left** and **top right** with tokens) and a modified version designed to resonate with the ritual and material heritage of the church (**bottom right**).

For both JigsAudio provotypes, simple silicon tokens cast with RFID tags were used (large enough to write a few words describing your audio story, and with a badge-pin on

the back). Both devices were kept in the church for the duration of the workshop period (hidden from view) with printed instructions and tokens (used and unused).

4.4.2. JigsAudio and Clumsy Affordances

Over the course of the seven workshops, several key insights emerged on the affordances of the two Jigsaudio provotypes. The first was the value of the device appearing integral to the church itself rather than as a curious object that 'stands out'. As one participant joked in reference to the original Jigsaudio: "Why is it so complicated? I started to think 'is this some kind of quantum time machine where you're communicating back with people in medieval times" (Participant 6). In contrast, the modified JigsAudio felt more connected to key characteristics of the church building as a place for thoughtful reflection and connection. Participants suggested that this resonance might be further developed, for example, through use of local materials or craftsmanship (e.g., for the device casing). The importance of recording personal histories using the device was clear (observation 2), particularly in how it helped contextualise the church in its community more fully. Over the course of the workshops, for example, participants used the provotypes to record a wide variety of personal histories, deep reflections, historical questions and home-spun philosophies. Here, value was ascribed to the act of contributing to something 'bigger than yourself' in the hope of helping future generations understand something of community life as it is today. In that varied use, the device revealed different user needs from an interaction (observation 3). For example, the time limit on recording was helpful for those wishing to record spontaneously, but challenging for others who 'speak freely when they get going'. For some, the stillness of the church helped them reflect but added pressure when wanting to be more spontaneous, whilst for others, having too much time to plan a response made it harder to say the 'right thing'. More can be carried out (through design, signposting and contextualisation) to help people receive the most out of the device according to what they want to 'put in'.

Working with the tokens over a number of weeks revealed a range of potential uses (observation 4), not only for documenting ideas but as a means of exchange, invitation, and a symbol of participation. Interestingly, this connects to a longer history of small badges and tokens in medieval life, where they might denote a completed pilgrimage, organisational membership or serve as votive offerings in church (for prayers that have been answered or favours requested) [55]. Bringing these observations together is the importance of differentiating between types of church audience, their needs and interests (observation 5). Currently there is only single 'visitor experience' at the church. Here, our second provotype helped explore how multiple routes to engagement might be created, differentiating between, for example, the needs of village residents wanting to share experiences (but who would not normally become involved), a CCT visitor who has in-depth knowledge to share (but who only has the visitor's book in which to leave their mark), and so on. Finally, the issue of device security, and how best to ensure it, was raised (observation 6). This is not only a practical issue, but also a social one with much wider implications. Security is always a concern in open churches, and here a tension emerges between a device that emphasises individual freedoms (open, easily-access, mobile, trust-based) or institutional control (anchored and access-regulated).

These reflections point to the potential clumsy affordances of JigsAudio according to our proposed trajectory for clumsiness (summarised in Table 4). The device offers a more personalised way for someone to express their own perspective, with those interactions both focused and moderated through the communal nature of the device, i.e., open to other users and used in public (T1: I→E). That shared-use and meaning is emphasised through the design-connection to the church building, its communal use in that location, and the use of tokens to form a collective record of community—all routes for affirming shared ownership and responsibility (T2: ↑E). Since the device can be made accessible to all who want to use it, there can be both greater openness in participation as well as 'why, how and when' contributions are made (T3: ↓H). Within this use-context, the device can create new

ways to express opinions, to be heard, to be invited to participate, to have influence on others, and so on, so broadening opportunities to be meaningfully involved (T4: ↓F). From this foundation, different 'flavors' of clumsiness can be proposed that create differentiated roles for different audiences in church activities:

**Table 4.** Summary of JigsAudio's provotype's clumsy affordances.

| | **T1: (I→E)** | **T2: (↑E)** | **T3: (↓H)** | **T4: (↓F)** |
|---|---|---|---|---|
| **Clumsy Affordances** | Individual contributions are encouraged in a social context | Device, tokens and building are both shared and communal | Different styles of participation are supported | Barriers to participation are lowered |

### 4.4.3. Four Flavors of Clumsiness

**The Individual User**: In individual use, the device offers creative freedom to record and exchange ideas, stories, proposals and observations, a driver for bringing new ideas into the lifecycle of community activities at the church (consistent with our clumsy trajectory). For some, the intimate bowl-like qualities of the device may help strengthen a focus on one's own thoughts and perspectives. The exchange of tokens (in private) would explicitly acknowledge that time *can* be taken for a response. Tokens might also be personalised, made reusable and have recording time limits removed—all introducing greater individual control in the style, scheduling, and content of interactions.

**Group Reflection**: At an event in the church, the device offers a way to draw out shared experiences, insight, and feedback in the form of short and spontaneous observations. By being shown how to use the device, and using it together, the mechanism for expanding the circle of people who know what the device is and how to use is built into church programming itself. The affordances of the device in this context for sharing and exchanging ideas emphasise a solidarity-building dimension (consistent with our clumsy trajectory). Recorded tokens might then be made available in the church itself through an installation, i.e., used to stage wider-community exchange and celebration.

**The Expert**: The value of deeper insight into lived experience and community history is clear as a necessary addition to the traditional historical and heritage knowledge commonly made accessible in churches. This greater diversity of expertise (consistent with our clumsy trajectory) might be better accessed through the device, an invitation (in connection with a bespoke token perhaps) to visit the church and record a longer, thoughtful, and carefully thought-through piece. This mode of use will require a degree of coordination and preparation. Recordings might be made for special occasions, in response to specific artefacts, events, festivities, and so on, and curated for access in the church.

**Reaching Out**: CCT churches have a small but regular stream of visitors interested in heritage and often looking to become more involved in some way (as volunteers). There is currently little to support at churches themselves for that engagement. With an estimated 1 in 7 signing a church's visitors book, the likelihood of engaging with a JigsAudio device unaided (however well sign-posted) is low. However, tokens offered as gifts with an invitation to return to the church at a later date (for example, to attend an event) opens a route to build that engagement. Tokens might be blank or have pre-recorded content (a prompt to use the device on the return visit when others are guaranteed to be there). Here, the long history of elaborate medieval badges, tokens and votive offerings could be a stimulus for token-design that people will want to take away with them and learn more about through a return visit.

## 5. Discussion

### 5.1. A Clumsy Placemaking Strategy

In this paper, we have demonstrated how a wicked/clumsy methodology can lead to new solutions that respond to the challenge faced by rural communities with empty churches. Clumsiness is defined in social terms, namely in how solutions draw on different

forms of social life to drive positive outcomes that can benefit more of the community. In the process, we have shown the potential of long history and slow technology to expand the space for clumsiness. The slow-history dimension has helped not only identify new sources of inspiration for pluralist church futures but also a reappraisal of the contemporary nature of the crisis at hand (pivoting from a crisis of modernity—in which the past is dispensed with to find 'new solutions'—to a clumsy continuation of an already changeful past). In parallel, the slow technology dimension has revealed how site-specific technologies might engage different audiences in generating, documenting and expanding participation in community activities centred on the church. Those elements, however, do not yet constitute a full, clumsy placemaking strategy, one that might deliver a programme of change and might be adopted by other communities wanting to transform their own historic place of worship through a socially driven (rather than infrastructure-led) regeneration approach.

One possible configuration of that strategy is outlined here (see Figure 6). We imagine that the five clumsy long history elements (Section 4.3) would form the core of the strategy, supporting 'perception change' activities that help uncover potentials and resonances within a community. Laying the foundation for thinking about their historic church differently, this critical exploration of the long-history of church adaptation aims to site those potentials at the church itself. In this core part of the strategy, the mechanism of 'Gifting and Redeeming Value' (derived from St Nicholas specifically) would become a standalone element, one relevant for all church sites. Exploring the local patron saint(s) (as we have carried out at Brockley) would serve only to deepen the value of this long history element through uncovering, for example, additional motivations and drivers within the Gifting and Redeeming model. The value of these long history elements can be demonstrated through our initial work at Brockley, where an initial cohort of over 20 project ideas across a wide range of domains were developed. These included 'Brockley TEDx'—re-launching a lecture series in the church to new audiences and more diverse speakers; 'wellbeing walks'—using local expertise to connect into Social Prescribing (community referral for wellbeing) and regional Ecosystem Services; 'suspended' coffees as a novel way to invest in a local farm shop, gain visibility and build social capital within the parish; a modern pilgrimage route—developing CCT's Champing (glamorous-church-camping) model to support 12 steps addiction recovery; 'Chilli Nights'—a community meal in the church around St Nicholas' day in defiance of the winter; and many more.

Building out from this core of 'perception change' activities, a number of milestones can be set for developing and delivering projects. Drawing from the learning of Empowering Design Practices [24], these milestones might include: a. 'mapping of local contexts' (e.g., around specific themes that have emerged or topics known to be important locally); b. 'activation of collaborators' (e.g., reaching out to others around themes of interest, also beyond the parish); c. 'communicating process and progress' (e.g., devising a communications plan, Vision and Mission statements, and simple decision-making structures for onboarding projects. Protocols for groups not bound by conventional committee structures have already been developed for this space [56]); d. 'building capacity' (e.g., providing training activities in digital and social media skills [57]); e. 'generating and testing new ideas' (e.g., within the framework of Gifting and Redeeming Value); f. 'mobilising support and funding' (e.g., where additional materials or professional services might amplify a Gift); and finally, g. the delivery and documentation of activities. The slow technology approach we have described in this paper would play a role in achieving a number of these milestones (see Section 4.4.3).

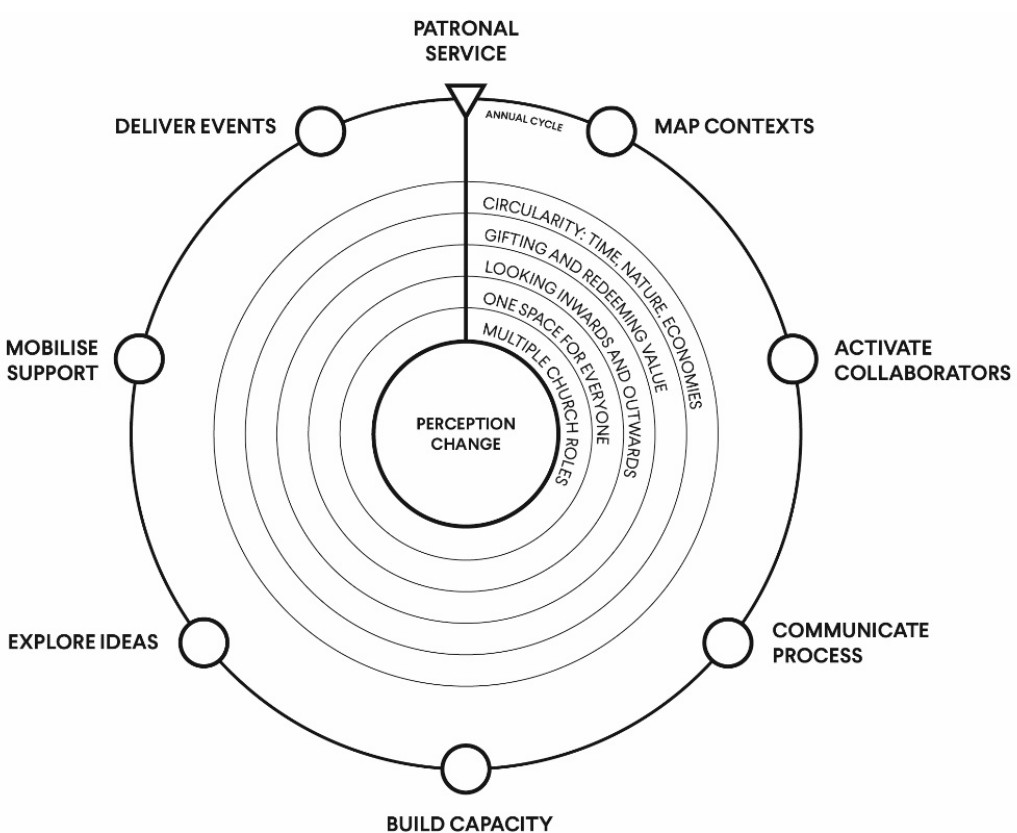

**Figure 6.** A clumsy placemaking strategy pivoting around a church's patronal day, incorporating a core of perception change activities and key milestones (see Section 5.1).

Our proposal of an annual programming cycle (Section 4.3.5), pivoting around the patronal day service, offers a ready-made structure to extend community engagement (Figure 6). The service itself would be an ideal moment to celebrate project impact, reflect on what has been achieved, and turn attention to the year ahead. Sequential annual cycles are a key opportunity to broaden participation and diversify activities (key to spreading risk, particularly where there's an over-dependence on a few key individuals), expand the pool of people interested in funding maintenance and repair projects, build participant motivation, confidence and capabilities, and widen revenue streams (with the long-term goal of reducing dependency on grants and public bodies). The nature of activities across cycles cannot be predicated in advance: some activities may establish an annual rhythm, others may remain episodic, and some may never repeat. An awareness of that changing composition of activities will help reveal resonances, opportunities, and emerging needs within the community. This process will, in-time, call for a reappraisal of the clumsy placemaking strategy itself, in part because the strategy (if successful) will change the demands being made of clumsiness. That might include, for example, closer (or altered) working relationships with other local organisations or the development of new decision-making structures and forms of governance. Further, whilst the long history and slow technology dimensions may have value now, they might only serve as transitional elements in establishing a new culture of church-use, one that comes to imagine and describe its own role differently to that at present.

*5.2. Reflection on Co-Design*

Whilst the value of design-led methods (such as Design Thinking and co-design) in the development of clumsy solutions has already been described [41], here we argue that insights gained through Wicked/Clumsy thinking may also benefit co-design itself. At a macro-level, Wicked/Clumsy thinking can challenge a tendency towards linear-thinking

between problem and solution definitions. Whilst 'non-linear' practices (from iteration through to embracing serendipity) are all part of the co-design toolbox, Wicked/Clumsy thinking explicitly draws attention to the way interventions sit within pre-existing and historically contingent contexts, and so impact the very conditions by which success might be measured. This calls for more than 'consequence scanning', forcing practitioners to ask how interventions can lay the grounds for their own repeated readjustment and renewal in the future. Similarly, it grounds 'futures thinking' in more than an appraisal of what is 'to hand' in the present, opening-up a deeper historical engagement that offers different kinds of potential for thinking about the future. Secondly, at a macro-level, Wicked/Clumsy thinking critically foregrounds 'the social' in co-design (detailed in our work through Cultural Theory). Engaging a plurality of voices and a deep understanding of the value of solidarity in collaborative work has always been a bedrock of co-design; Cultural Theory, however, supports a critical engagement with people's construction of their own social realities and how those constructions are embedded within local institutional structures and practices. The route towards building solidarity through co-design can, therefore, be paved in different ways appropriate to who is participating. That deeply projecting social lens has an important secondary effect in directing attention to 'the social' in the substance of co-design itself: a good example is an expansion of historical and heritage dimensions in this field of work from 'designerly' subjects (buildings, objects and artefacts) to include social histories and social heritage. Finally, at a micro-level, Wicked/Clumsy thinking lays out a research agenda for identifying and then implementing a trajectory towards clumsiness. The three-stage agenda demonstrated here could be used at other community sites to uncover their own conditions for clumsiness. We argue that the potential for co-design—projected through Wicked/Clumsy thinking—to impact communities with under-used churches is considerable. The emergence of viable policy instruments, as demonstrated by the Taylor Review pilot, now set the agenda for this work at scale.

**Author Contributions:** Conceptualization, T.J.S., M.A., S.B., T.M., S.M., E.M. and A.W.; methodology, T.J.S.; software, A.W. and T.M.; formal analysis, T.J.S. and S.M.; investigation, T.J.S., M.A. and E.M.; resources, A.W.; data curation, T.J.S.; writing—original draft preparation, T.J.S.; writing—review and editing, M.A., S.B., T.M., S.M. and A.W.; visualisation, A.W.; supervision, T.J.S.; project administration, T.J.S. and T.M.; funding acquisition, T.J.S., M.A., S.B., T.M. and A.W. All authors have read and agreed to the published version of the manuscript.

**Funding:** This research was funded by the Brigstow Institute (seedcorn funding) at the University of Bristol.

**Institutional Review Board Statement:** Ethics approval was obtained from two universities for different stages of the project. Ethical approval for the qualitative interview data was obtained from the University of the West of England Faculty Research Ethics Committee. Written information about the study was circulated to research participants in advance. Written informed consent was obtained from all participants immediately prior to each interview. Ethical approval for the workshops and co-production of outputs was obtained from the UoB Arts faculty Research Ethics Committee.

**Informed Consent Statement:** Informed consent was obtained from all subjects involved in the study.

**Data Availability Statement:** The interview data presented in this study are not publicly available due to ethical and privacy restrictions.

**Acknowledgments:** We would like to thank The Churches Conservation Trust and the community at Brockley for their participation in this project and dedication to seeing it through. Also with thanks to Harri Hudspith, Lauren Cole and Sarah Wordsworth for their support in exploring rural medieval life.

**Conflicts of Interest:** The authors declare no conflict of interest. The funders had no role in the design of the study; in the collection, analyses, or interpretation of data; in the writing of the manuscript, or in the decision to publish the results.

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
