# Peer review of "Medievals and Moderns in Conversation: Co-Designing Creative Futures for Underused Historic Churches in Rural Communities"

_mti, doi:10.3390/mti6050040_

Round 1

Reviewer 1 Report

The article discusses the effectiveness of co-design in finding appropriate solutions to the complex problem of under-use of historic churches in rural areas. The proposed interpretations are continuity with the past of places and communities, as well as "slow technologies", through new forms of longer-term learning, engagement and reflection in place. The authors adopt the Wicked/Clumsy approach to co-design processes.
The illustration of the background is well organised, the methodology adopted is rigorous and clearly explained. The results achieved provide an original contribution to the scientific debate. The references are appropriate.

Author Response

We thank reviewer 1 for their comments and are pleased that our paper provides “an original contribution.”

Reviewer 2 Report

The paper helps, using Wicked /Clumsy co-design methodology,  to imagine new futures for communities and their historic churches today.

The paper is interesting and well written.

There are a few things that could be made clearer:

  • better explain the workshops, aims, and outcomes;
  • why provotype (a provocative prototype)?
  • In the Discussion section, it is written (lines 939 and following), "We imagine that the five clumsy long-history elements (section 4.3) would form the first phase of the strategy,...". Could you give more information about the 4.3. Finding Clumsiness in Long-history (Stage 3) - the first phase of the strategy and their relationship?

There are small formatting errors in the numbering of tables and paragraphs

  • paragraph 2.3 is missing
  • some tables leave the page and are cut off 
  • reference to table 2 line 640: probably refers to table 3

I suggest to accept the paper with minor revision

Author Response

We thank reviewer 2 for their constructive comments, which we have used to improve our paper as described in our point by point response below:

  • Workshops in relation to the whole: We have improved the way we describe the relationship between the three stages of the project (3.3), improved the diagram that shows how three stages link together (Figure 3) and improved the headings of the three stages to reflect that connection. We think this makes it clearer how the workshops serve the dual purpose of finding clumsiness in long history and slow technology whilst also embodying in their own design (necessarily) the types of social interaction that can help those conversations happen (i.e. already putting the trajectory towards clumsiness into action).
  • Why Provotypes: We have clarified that both slow technology devices (the original JigsAudio and the modified version) are provocative prototypes, i.e. examples of how slow technology might be used in conversations about church futures, a way to stimulate (provoke) discussion on slow technology’s particular (potentially clumsy) affordances in that role (3.3). We have also made it clearer what the original Jigsaudio was used for (in a slow technology role) and where more information can be found out about that role.
  • The final placemaking model in the discussion: We have now re-written and clarified this section, describing how the long history elements in 4.3 form a collection of perception change activities that lay the ground that form the core of the place-making model, that model incorporating a series of milestones that can then be delivered to bring change into effect. A new accompanying figure (Figure 6) has been developed to describe the relationships between these parts and show the model as a whole.

Formatting errors: These have now all been corrected

Reviewer 3 Report

Medievals and Moderns in Conversation: Co-designing Creative Futures for Under-used Historic  Churches in Rural Communities

 This paper bridges the medieval historical memory of England church with the scope of today human communicates. Very well presented and organized, with very appropriate references and a clear presentation of a case study - St Nicholas’ church. But it’s not a paper on empirical and single focus. It goes further from St Nicholas’ church and gives real solutions based In the search for a Clumsy solution, with key sources of inspiration: ‘long-45 history’ and ‘slow technology’, or «Weakness -    to a  Clumsy Trajectory».

This collective paper - issue of a  UK project – is a highlight issue on transdicisplinary academic work – conceptual theoretical framework (cultural theory, e.g), and an impressive field work with local volunteers.  Placed on a local territory – from the medieval parish times to the present (with the Covid 19 sensibility) gives to the lecture and to the general and academic public very interesting solutions on church and local communities and cultural heritage and human heritage and ways of life, today. The expression: «Rural life is subject to change, not a driver of  change (expressly in contrast to life in urban areas)».(Line 502) gives the scientific core research of the project that was  was funded by the Brigstow Institute (seedcorn funding) at the University 1030 of Bristol.

Author Response

We thank reviewer 3 for their comments and are pleased that they consider it “highlights  transdicisplinary academic work” and provides a conceptual theoretical framework and “impressive field work with local volunteers”.